# Interactome Analysis and Docking Sites of MutS Homologs Reveal New Physiological Roles in *Arabidopsis thaliana*

**DOI:** 10.3390/molecules24132493

**Published:** 2019-07-08

**Authors:** Mohamed Ragab AbdelGawwad, Aida Marić, Abdullah Ahmed Al-Ghamdi, Ashraf A. Hatamleh

**Affiliations:** 1Genetics and Bioengineering, Faculty of Engineering and Natural Sciences, International University of Sarajevo, 71210 Sarajevo, Bosnia and Herzegovina; 2Centre for Research in Agricultural Genomics, UAB-Edifici CRAG, Cerdanyola, 08193 Barcelona, Spain; 3Department of Botany and Microbiology, College of Sciences, King Saud University, Riyadh 11451, Saudi Arabia

**Keywords:** DNA mismatch repair, MSH, docking site, interactome

## Abstract

Due to their sedentary lifestyle, plants are constantly exposed to different stress stimuli. Stress comes in variety of forms where factors like radiation, free radicals, “replication errors, polymerase slippage”, and chemical mutagens result in genotoxic or cytotoxic damage. In order to face “the base oxidation or DNA replication stress”, plants have developed many sophisticated mechanisms. One of them is the DNA mismatch repair (MMR) pathway. The main part of the MMR is the MutS homologue (MSH) protein family. The genome of *Arabidopsis thaliana* encodes at least seven homologues of the MSH family: AtMSH1, AtMSH2, AtMSH3, AtMSH4, AtMSH5, AtMSH6, and AtMSH7. Despite their importance, the functions of AtMSH homologs have not been investigated. In this work, bioinformatics tools were used to obtain a better understanding of MSH-mediated DNA repair mechanisms in *Arabidopsis thaliana* and to understand the additional biological roles of AtMSH family members. In silico analysis, including phylogeny tracking, prediction of 3D structure, interactome analysis, and docking site prediction, suggested interactions with proteins were important for physiological development of *A. thaliana*. The MSH homologs extensively interacted with both TIL1 and TIL2 (DNA polymerase epsilon catalytic subunit), proteins involved in cell fate determination during plant embryogenesis and involved in flowering time repression. Additionally, interactions with the RECQ protein family (helicase enzymes) and proteins of nucleotide excision repair pathway were detected. Taken together, the results presented here confirm the important role of AtMSH proteins in mismatch repair and suggest important new physiological roles.

## 1. Introduction

Living organisms are exposed to different damaging factors at all times. Therefore, maintenance of genome stability and integrity is one of the key roles of a cell. DNA damaging factors jeopardize the integrity of the DNA and can come from endogenous and exogenous sources [1]. Just as there are many damaging factors, organisms developed diverse pathways to fight against deleterious DNA damage and to retain genomic stability [2,3,4,5,6,7,8]. Plants are in special need of effective DNA repair machinery. They do not have a continuous germ line, but meristematic cells give rise to the gametes. These meristematic cells divide and potentially accumulate mutations during the lifetime of plants. Without repair, these mutations will be passed on to the next generation. While DNA repair pathways are well understood in yeast and mammalians, our knowledge in plants falls farbehind. Therefore, there is a need for more research that will shed additional light on this interesting and powerful part of plant genomes. This work focuses on the mismatch repair (MMR) pathway in *Arabidopsis thaliana*. Mismatch repair MMR is post-replicative DNA repair machinery. It is able to recognize non-Watson–Crick pairing as well as insertion/deletion loops (IDLs) [9]. Additionally, MMR has several other functions; it controls homologous recombination (HR) and most probably prevents synapse formation between divergent sequences [9]. Together with DNA polymerases and exonucleolytic proofreading, MMR keeps high fidelity of DNA with only one mispair every 10^10^ bases [10]. The mismatch creates a nick in the DNA helix and is recognized by MutS or its eukaryotic counterparts—MutS homolog (MSH) proteins. The MSH recruits downstream proteins that make a nick in the new strand. Exonuclease is then recruited to cut out part of the DNA strand surrounding the mismatch. The gap is finally filled in by DNA polymerase and sealed with DNA ligase. On the other hand, since proofreading exonucleases have limited capabilities, IDLs will mostly be repaired by MMR machinery. Besides its role in post-replicative point mutation repair, MMR plays an important dual role in homologous recombination. First, MMR recognizes mismatches in recombination intermediates, but on the other hand, MMR is able to prevent recombination between diverged sequences and excessive exchange of their genetic material [11,12]. Extensive duplication events enabled MSH proteins of MMR to specialize and recognize a variety of mismatches [13,14]. The MSH proteins are present throughout all kingdoms of life, suggesting conservation of MMR through the evolution [15]. The versatility of MSH proteins in eukaryotes enables MMR to recognize a surprising amount of different mutations. This work will focus on MutS homologue (MSH) proteins in plants. *Arabidopsis thaliana* encodes seven MSH homologs. AtMSH1 is thought to be the only non-nucleus-based MSH protein. It is dually targeted to mitochondria and chloroplast and plays a very important role in maintaining the stability of their genomes [16]. AtMSH1 mutants show anincreasein the reorganization of the mitochondrial genome and result in decreased abiotic stress response, fluctuation in growth dynamics, extended flowering and maturity, and reduced heat tolerance and sterility [17,18]. AtMSH2 protein is involved in the initiation of MMR and recognition of mismatch. Besides, it is involved in the control of DNA HR and prevents recombination of divergent strands [19]. Additionally, AtMSH2 is part of the repair pathway of UV-induced DNA damage [20]. AtMSH3 is an MMR protein that works in conjunction with AtMSH2. Together, they form a MutS beta heterodimer that recognizes damage and initiates repair of DNA loops of different sizes [14]. AtMSH4 is a somewhat different member of the plant MSH family which is not directly involved in MMR. Instead, MSH4 regulates meiotic recombination and keeps it at a normal level. AtMSH4 is only present in floral tissues, which is in line with its role in reproduction [21]. AtMSH5 works in association with AtMSH4. It is expressed in flower tissue and promotes proper segregation during chiasma formation in prophase I. AtMSH5 mutation leads to serious fertility reduction [22,23]. AtMSH6 forms a MutS alpha heterodimer with MSH2 that recognizes base–base mismatches and short (trinucleotide) IDLs. AtMSH7 is a plant-specific MSH protein. With AtMSH2, it forms a MutS gamma heterodimer that recognizes only T/G mismatch and initiates mismatch repair. Due to the complexity of the topic and limited amount of information available on AtMSH proteins, the aim of this work is to shed additional light on the function of AtMSHs, leaning on the predicted structure, detailed interactome analysis of the proteins, and docking prediction.

## 2. Results

### 2.1. Multiple Sequence Alignment

ClustalOmega aligned sequences of seven AtMSH proteins and retrieved results are shown in Appendix A. Residues were colored based on their physicochemical properties (small and hydrophobic residues are in red; acidic in blue; basic in magenta; and hydroxyl, sulfhydryl, and amine in green). These results are quantified and available in the form of a percent identity matrix in Table 1. Multiple sequence alignment MSA showed highest identity between MSH6–MSH7. This is in line with previous research that suggested MSH7 diverged from MSH6 [14]. The second highest scoring pair was the MSH2–MSH3 dimer. The highest divergence was noticed for the MSH1 protein. This is in line with the expected results since MSH1 is a mitochondrial protein.

### 2.2. Phylogenetic Profile Rendering

Results retrieved from Phylogeny.fr for *A. thaliana* MSH proteins are visible in cladogram (Figure 1). During the long course of evolution, *MutS* genes of endosymbiotic bacteria gave rise to a specialized group of *MSH* genes. This was achieved through multiple duplication events [24]. The phylogenetic tree supports the theory that *MSH1* was originally a mitochondrial gene. This will be further elaborated in the discussion. 

### 2.3. Protein 3D Structure Prediction and Refinement

Three-dimensional modeling is a cornerstone of modern structural biology. Determination of the protein structure is the most important step towards the determination of its function, determining possible ligands and docking sites, and finding conserved motifs and domains. The structures here are the result of a bioinformatics approach and are based on homology modeling. The results of the 3D structure prediction are given in Figure 2. 

### 2.4. Protein 3D Structure Validation

Both experimental and in silico models of the 3D structure have to be validated before being named acceptable. Bioinformatic tools use different references to validate a model; measuring bond distances, bond energy, torsion angles, B-factor, free energy of the molecule, etc. The results of the Ramachandran plot assessment for each AtMSH protein model are visible in Figure 2. Further validation was done in Model Quality Assessment Programs (MQAPs) such as PROCHECK, which certify the stereochemical properties of the model and use the free energy scoring tool dDFIRE to assess energy functions by ab initio refolding of fully unfolded terminal segments with secondary structures while keeping the rest of the proteins fixed in their native conformations [25]. Summary of results from all the validation tools is shown in Table 2. The tools render the MSH models as reliable. 

### 2.5. Protein Domain Identification

Domains are conserved regions of protein that can be a strong indicator of its function. Conserved regions of MSH proteins, as detected by SMART (Simple Modular Architecture Research Tool), are listed in Table 3. All proteins share a MUTSac domain [26]. This is the ATPase domain of the MSH proteins located at the C-terminal [27]. Although detailed information is not available from the eukaryotic MUTSac domain; the prokaryotic model suggests that only one monomer of the MSH dimer binds ADP through the MUTSac domain. Mismatch recognition initiates ATP binding which results in conformational change of the dimer and its movement along the DNA. Another domain that was present in all MSH protein except mitochondrial MSH1 is MUTSd.

This is a DNA-binding domain of MutSfamily, anda core domain made up of two subdomains that bind the DNA as levers. This domain is homologous to domain III of MutS in *Thermusaquaticus* [28]. Both MutS_I and MutS_II domains were identified by Pfam. They are homologous to domain II of *Thermusaquaticus* [29]. These domains functionally resemblethe RNase H domain that is responsible for RNA digestion and related to reverse transcriptase action. Similarly, MutS_II corresponds to domain II of MutS of *Thermusaquaticus* and is involved in DNA binding by MutS. Of all MSH homologues in *A. thaliana*, GIY-YIG domain is present only in MSH1. This is a catalytic domain present at the N-terminal of endonuclease [30] and its connection to DNA repair has already been inferred [31]. Finally, the TUDOR domain is present only in MSH6.The proteins that contain the Tudor domain are known as histone modification and categorized as chromatin remodeling proteins. Gene expression and DNA replication are greatly affected by histone modifications and chromatin remodeling, but how these processes are incorporated has not been fully investigated. It obvious that TUDOR domain proteins are development regulators carrying out functions that are not disclosed in plants [32].

### 2.6. Interactome Analysis

Before interactome analysis, proteins were assessed for solvent accessibility using the Protein Predict server. The results showed high solvent accessibility of all homologs which was an indication that we can expect extensive interactions and interactome profiles. The interactome analysis was a keystone of this work. It provided valuable information about proteins and protein families that interact with MSH homologues of *A. thaliana* (Table 4; Appendix A). All AtMSH homologues interact with thethree core proteins MLH1, MLH3, and PMS1 (Postmeiotic Segregation 1). This wasexpected since these are the plant eukaryotic counterparts of bacterial MutL and play important roles in MMR. All the AtMSH proteins, with the exception of AtMSH4, interacted with PCNA1 and PCNA2 (proliferating cell nuclear antigen), which play important roles in DNA replication as sliding clamps that enable elongation of leading strands [33]. Four out of seven homologues (MSH1, MSH2, MSH6, and MSH7) interact with TIL1 and/or TIL2—proteins with important physiological roles in plant growth and development. Extensive interactions were noticed between MSH homologues, RECQSIM and ERCC1. These proteins are an important part of other DNA damage repair pathways. Seven out of ten MSH4 interactors were not seen in any other homologue, which is an indication of a specific role of this protein. MSH5 extensively interacted with DNA helicases (RECQ4A, RECQSIM, RecQI3, RECQI1, RECQ4B).

### 2.7. Protein Subcellular Localization

Organelles are in charge of different cellular processes and hold different sets of proteins. Therefore, protein localization represents an important step in deciphering protein function, but is also suggested to be key to functional diversity [34]. Localization of AtMSH proteins is given in Table 5.

### 2.8. Docking Site Prediction

Proteins develop their functionality through interactions with other macromolecules (DNA, RNA, other proteins, etc.). Therefore, understanding protein–protein interaction is crucial for elucidation of its function and the analysis of the whole proteome. Results obtained from ClusPro and SPIDDER were in line with each other. Each AtMSH protein was docked against its most interesting interactors. The results from ClusProareare are shown in Appendix A. In order to confirm the results, the docking was done in SPPIDER http://sppider.cchmc.org/ and the results coincided with the ClusPro analysis and were accompanied by tables indicating active sites of AtMSH protein and its interactor (Appendix A).

## 3. Discussion

DNA, just like all organic molecules, can undergo chemical changes. However, structural changes on DNA have much larger and far-reaching consequences. DNA mutations can arise as result of DNA replication slips or spontaneous chemical changes stemming from exposure to different damaging factors. Therefore, cells had to evolve mechanisms to cope with these damages. One of them is the mismatch repair pathway. Most important proteins of this MMR are MutS homologs (MSHs). Evolutionary conservation of the MMR pathway and MSH orthologs in the plant and animal kingdoms, being higher in comparison to bacterial counterparts, allows us to transfer knowledge from *A. thaliana* to animals. Plants carry seven homologs of MSH, compared to five homologs in humans and six in yeast.Therefore, it was very interesting to check for functions of plant MSH homologs in MMR, estimate potential redundancy in their role, and look for new avenues in which they function.

Although the majority of MutS homologs belong to the same protein family, a certain degree of functional diversity among MSH proteins was observed. Phylogenetic analysis of plant MutS homologs confirmed this functional specification. This is especially striking in the example of MSH6 and MSH7 which diverged recently but have different functions. It would be of great importance to find out which amino acids are responsible for this functional specification. It is of great significance to look at genes and their occurrence from an evolutionary perspective. This is why the first step was to look at MSA and a phylogenetic tree of MSH proteins. MSA revealed a ~200 amino acid long conserved region at the C-terminal that suggested the core domain of the AtMSH protein family and a common ancestor of these proteins. This is in line with previous studies [35]. As indicated by Culligan et al.,the first duplication event enabled one copy to encode the mitochondrial MSH1 protein and the other copy gave rise to a diversified MSH family. The secondnuclear duplication gave rise to ancestors of (i) MSH6 and MSH7 and (ii) MSH2, MSH3, MSH4, and MSH5. Further duplication and specialization enabled MSH6, the MSH7 ancestor, to form subfamilies of MSH6 and MSH7. On the other hand, duplication gave rise to the meiosis-specific MSH4 subfamily and ancestor of MSH2, MSH3, and MSH5. By further duplication, the later three diverged into separate genes. This diversification was followed by specialization in function. Further analysis showed that the conserved region identified as a core domain by MSA is a MUTSac domain. Some regions are conserved only in subfamilies of MSHs, so they can confer specific functions to these proteins. Pfam:MutS_I (PF01624) is a domain with unknown function, but judging by its presence in all MSH proteins except MSH4 and MSH5, which does not have DNA mismatch binding ability, it is reasonable to hypothesize that MutS_I is a DNA binding domain, as suggested by studies in corresponding domains of *Thermusaquaticus.* The interactome of MSH proteins shown here shows that these proteins have crucial roles in plants: (i) they maintain the stability of nuclear and organellar DNA and (ii) control numerous physiological processes. This is not the first time that proteins of DNA repair mechanisms were found to influence physiological characteristics of plants [36].

Subcellular localization indicates that MSH homologs are predominantly placed in the nucleus. The only exception is MSH1, which is localized in mitochondria and chloroplast. This was proved to be essential for substoichiometric shifting in plant mitochondria, stability of plastid genome, and consequently for plant growth, through interactions described below [37]. MLH1–MLH3 with MSH homologs suggests similar roles for plant homologues. It is important to note that MLH1 mutants in *A. thaliana* exhibit reduced fertility [38,39]. This is another important physiological trait directly influenced by MSH homologs. Proposed mechanisms of MLH1 function and its interaction with MSH homologs are certainly worth investigating further in plants.MSH homologs, with the exception of MSH4, extensively interact with replication factors PCNA1 and PCNA2. This is another proof of their importance for maintenance of DNA integrity, as these proteins have already been extensively studied in relation to DNA repair [35]. Additionally, these proteins have been correlated with the control of shoot differentiation and meristem organization, indicating another venue influenced by MSH proteins [40]. MSH1 interacts extensively with proteins involved in replication and recombination. The results shown here support the hypothesis that replication initiation is mediated by recombination, which would explain interaction with both groups of proteins.

One of the physiological roles influenced by MSH homologs includes plant growth, controlled by the mitochondrial genome rearrangements through the interaction of MSH1–RECA3 proteins. RECA3 is a protein involved in recombination and strand transfer activity, whose mutants were found to influence plant growth, leaf variegation, and altered leaf morphology [41,42,43]. MSH1–RECA3 interaction is supported by the same subcellular localization and involvement in the same substoichiometric shifting process. RECA3 interacts with MSH1 through the AAA domain, but it is possible to see a hole in the docking site (Appendix A). Using the results obtained in the BindN program, which identified DNA-binding residues in the interacting site (Arg212, Ser213, Arg214, Gly216, Ser94, Thr95), we can hypothesize that it is the area where DNA binds. Extensive interaction of MSH (MSH1, MSH2, MSH6 and MSH7) proteins was observed in relation to DNA polymerase epsilon catalytic subunit A (TIL1/POL2a/TILTED1) and/or DNA polymerase epsilon catalytic subunit B (TIL2/POL2b/TILTED2) [44]. Partial interaction through domains (MUTSac, Pfam:MutS_II, Pfam:MutS_I) was supported by interactions through electrostatic charges. TIL1 and TIL2 proteins alter root and shoot development, repress flowering, homologous recombination, abscisic acid signaling, and cell cycling [41,43]. This way, MSH homologs could be involved in the control of physiological characteristics of plants. Thus far, it was discovered that TIL1 mutants (abo4-1) have higher rates of homologous recombination and display pleiotropic defects in both vegetative and reproductive development, but the mechanism behind this was not investigated [45]. Another interesting protein found in the interactome was RECQSIM. Important information about it come from the work of Bagherieh-Najjar et al. who indicated that RECQSIM has a role in DNA repair and recombination, but they did not propose the mechanism behind this repair [46]. MSH2, MSH3, MSH5, MSH6, and MSH7 interactions with RECQSIM (through MUTSac, Pfam:MutS_II, Pfam:MutS_I domains, mostly supported by electrostatic charges),which is indicated here, proposes a possible model by which RECQSIM can contribute to DNA repair and genome stability, and consequently, influence plant growth and development. MSH5 interacts with six members of the RECQ family. This extensive MSH5–RECQ interaction indicates the important role of the RECQ family for plant DNA stability and fertility. ERCC1 and UVH1 are proteins involved in nucleotide excision repair and in mitotic homologous recombination [47,48]. AtMSH homologs were found to interact with AtRAD1 and AtRAD10 at the highest confidence value. AtMSH2 interacts with AtERCC1 and AtUVH1 through theC-terminal domain MUTSac between the 659th and 855th residue; while AtMSH3 interacts with UVH1 through MUTSd, a DNA-binding domain. Therefore, we can assume that MutSβ (MSH2–MSH3 heterodimer) is responsible for HR. This is in line with research done in yeast [49].

MSH4 is a special member of the MSH protein family. It has a role exclusively in meiotic recombination and not in MMR. It has a special interactome and different domain profile, where MSH4 shares only MUTSac (ATPase domain) and MUTSd (DNA-binding domain) with other MSH proteins and does not contain any MMR-related domains. As meiosis-specific protein, MSH4 interacts with AtSPO11-1 and AtSPO11-2 proteins through the MUTSac domain. AtSPO11-1 and AtSPO11-2 are components of topoisomerase 6, responsible for formation of DSBs [50]. Localization of MSH5 is dependent on the occurrence of MSH4; therefore, they do not have a redundant role [22]. Instead, MSH5 plays an important role in stabilizing chiasma during meiosis, and directly influences the fertility of the plant. MSH5 extensively interacts with RECQ homologs. It was found in *E. coli* and *S. cerevisiae*, and the mutation in RECQ leads to increased levels of recombination. This functional link to recombination and their interaction with AtMSH5 is a sign that they could have the same function in plants. What the exact mechanism of interaction between MSH5 and RECQ is a line of research worth exploring further. AtRECQ2 disrupts D-loops and prevents non-productive recombination events or channel repair pathways into non-productive recombination. Knowing that AtMSH5 is involved in meiosis regulation, AtMSH5–AtRECQ2 interaction is potentially very important for the fertility of plants, but to our knowledge, this has not been explored yet.

## 4. Materials and Methods

### 4.1. Sequences Retrieving and Multiple Sequence Alignment MSA

The amino acid sequences of seven AtMSH homologs were first retrievedfrom the National Center for Biotechnology Information (NCBI) [51] and the Arabidopsis Information Resource (TAIR) [52] (Table 6). Obtained sequences were aligned using the Clustal Omega tool [53,54,55,56].

### 4.2. Phylogenetic Profile Rendering

Sequences of AtMSH proteins were submitted for phylogenetic analysis in Phylogeny.fr. One Click mode was used here to construct a phylogenetic tree of AtMSH homologues using the neighbor joining method [57,58].

### 4.3. Protein 3D Structure Prediction and Refinement

The 3D structure of MSH homologues in *A. thaliana* has not been determined yet. Therefore, the homology modeling method was used to predict their 3D structure. Amino acid sequences of MSH proteins were submitted to Phyre2 (Protein Homology/analogY Recognition Engine V 2.0) portal [59]. In order to obtain structures closer to native state, the .pdb files retrieved from Phyre2 were refined using the protein structure refinement server 3Drefine [60,61]. 3Drefine is a free web server that brings the structure closer to a native state. It uses a two-step approach. First, hydrogen bonds are optimized, and second, energy at the atomic level is minimized. The critical assessment of techniques for protein structure prediction (CASP), which is used as the gold standard for the assessment of bioinformatics tools, recognized 3Drefine as a tool that brings structural improvement at the global and local levels of protein structure. Three-dimensional visualization of the protein surface was done using PyMOL software [62]. Additional visualization of 3D structure was done in DeepView-Swiss-PdbViewer available at ExPASy Bioinformatics Resource Portal [63]. DeepView is a powerful tool for macromolecular modeling that enables visualization of electrostatic potentials of proteins [64].

### 4.4. Protein 3D Structure Validation

After 3D structures were predicted as described above, these models were validated using several tools. First, the RAMPAGE tool was used for assessment of the Ramachandran plot [65]. A Ramachandran plot aligns backbone angles ψ (C–Cα bond) and φ (C–N bond) [66]. This is arguably the best assessment of the 3D structure prediction.

### 4.5. Protein Domain Identification

In order to identify functional domains of AtMSH proteins, SMART (Simple Modular Architecture Research Tool); Normal mode was used [67,68]. Besides SMART default HMMER search that uses hidden Markov models, Pfam domains were included.

### 4.6. Interactome Analysis

Following domain identification, proteins underwent interactome analysis. This was done in order to identify which proteins interact with AtMSH proteins. For interactome analysis, a STRING (functional protein association network) database was used [69]. STRING integrates information scattered over multiple databases in order to report on physical and functional protein–protein interactions. Protein sequences were submitted to STRING and parameters were set to show 10 interactors of highest confidence (>0.900).

### 4.7. Protein Localization

Localization is an important indication of protein function and biological interaction. Subcellular localization was done using online tools and literature. The PSI-predictor (Plant Subcellular Localization integrative predictor) was exploited [70]. It combines group voting and a neural network to integrate data from 11 independent predictors and outperforms all of them individually. For AtMSHproteins that were localized to the nucleus, further characterization was done in order to check in which part of the nucleus they are localized. This was done using Nuc-Ploc: predicting protein subnuclear localization [71].

### 4.8. Docking Site Prediction

The AtMSH homologs and their corresponding interactorswere submitted to docking site prediction tools, in order to visualize regions responsible for interaction. Two docking tools were used in this study: ClusPro and SPPIDER [72,73,74,75]. In 2004, when it was published, ClusPro was first a completely automated program for computational protein docking. ClusPro creates over 70,000 possible conformations, evaluates complexes, and selects ones with the highest surface complementarities and optimal electrostatic characteristics. ClusPro showed very good results in the Critical Assessment of Prediction of Interactions (CAPRI) and confirmed that cluster size-based ranking is reliable for identification of near-native conformations [76]. Solvent accessibility-based Protein–Protein Interface identification and Recognition (SPPIDER) offers a user-friendly website and is able to detect protein interfaces in two ways [77]; it has a different approach compared to other interface prediction tools. It uses relative solvent accessibility (RSA) as a reference point and calculates the RSA of an amino acid in (a) predicted and (b) unbound state. Here, RSA loss was set to at least 4% after the formation of the complex.

## Figures and Tables

**Figure 1 molecules-24-02493-f001:**
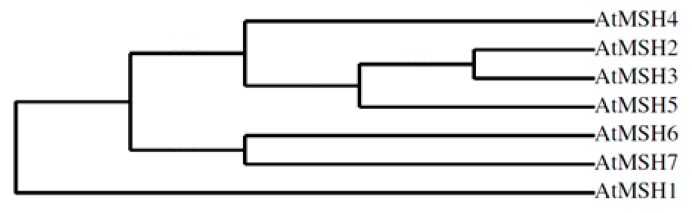
Cladogramphylogenetic tree representing the evolutionary relationships of AtMSH proteins.

**Figure 2 molecules-24-02493-f002:**
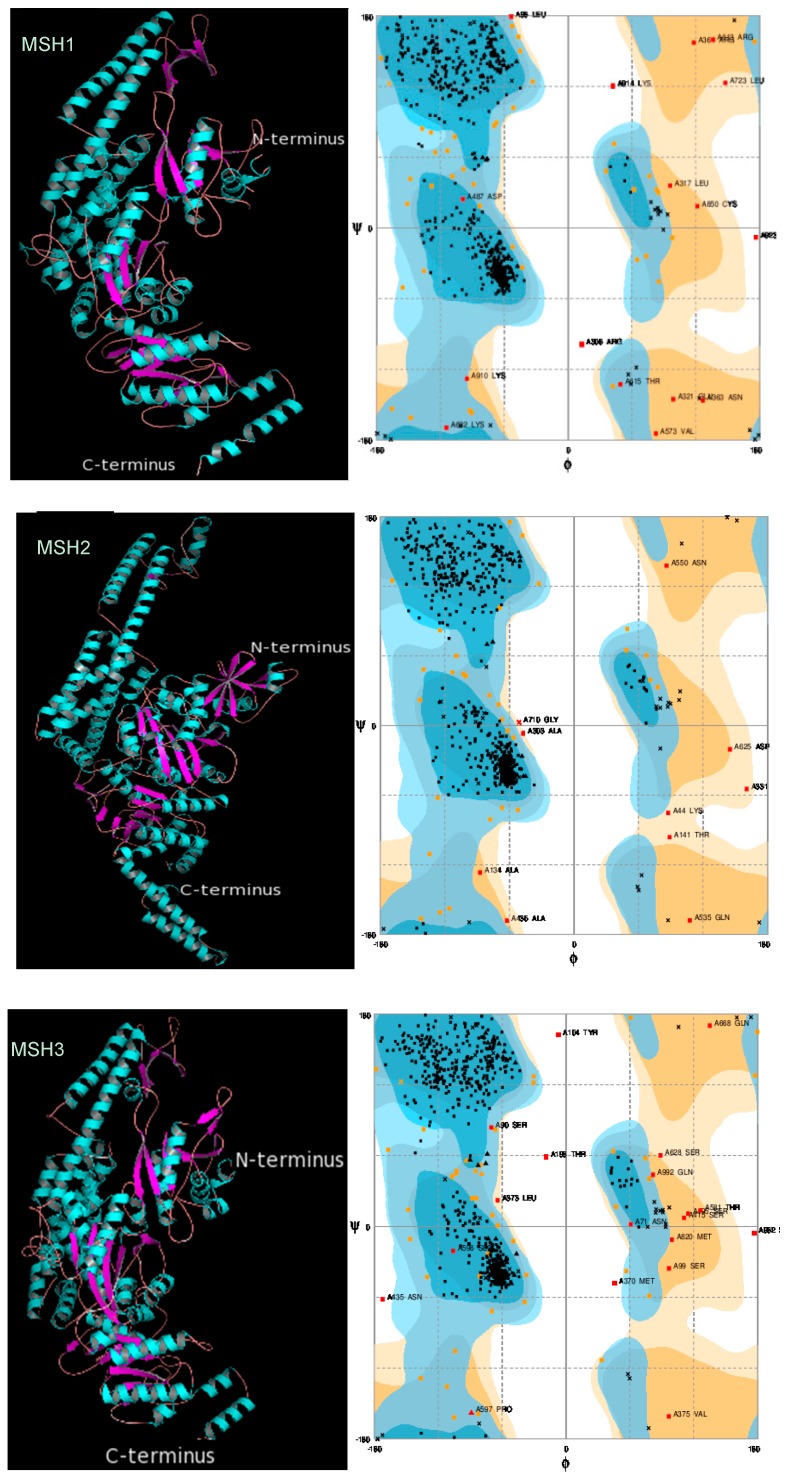
MSH1-MSH7 proteins’ 3D structure prediction visualized by PyMOL (left) and validated by Ramachandran plot (right). N-terminus and C-terminus are indicated.

**Table 1 molecules-24-02493-t001:** The identity matrix percent of similarities among AtMSH proteins.

Protein	Percent Identity						
	MSH1	MSH4	MSH5	MSH6	MSH7	MSH2	MSH3
MSH1	100.00	19.08	18.03	22.86	21.49	19.09	20.36
MSH4	19.08	100.00	22.51	24.53	25.29	24.56	22.88
MSH5	18.03	22.51	100.00	23.60	22.33	22.93	24.16
MSH6	22.86	24.53	23.60	100.00	33.30	23.95	26.30
MSH7	21.49	25.29	22.33	33.30	100.00	25.37	25.56
MSH2	19.09	24.56	22.93	23.95	25.37	100.00	27.93
MSH3	20.36	22.88	24.16	26.30	25.56	27.93	100.00

**Table 2 molecules-24-02493-t002:** 3D structure prediction verification tools of MSH proteins.

Protein	RAMPAGE (Residues in Allowed Region)	PROCHECK (G-Factor)	dDFIRE
**MSH1**	98.1%	−0.28	−1755.42
**MSH2**	98.9%	−0.05	−2105.67
**MSH3**	98.0%	−0.18	−2052.20
**MSH4**	98.1%	−0.16	−1753.06
**MSH5**	98.7%	−0.23	−1797.04
**MSH6**	98.1%	−0.18	−2093.36
**MSH7**	98.3%	−0.19	−1841.97

**Table 3 molecules-24-02493-t003:** Domains of MSH proteins.

Domain and Accession		Protein						
		AtMSH1	AtMSH2	AtMSH3	AtMSH4	AtMSH5	AtMSH6	AtMSH7
**MUTSac (SM000534)**	Start	761	659	810	546	562	1076	846
	End	947	855	1006	733	757	1268	1043
**MUTSd (SM000533)**	Start	-	314	440	190	211	716	573
	End	-	642	793	531	547	1056	822
**Pfam:MutS_I (PF01624)**	Start	125	22	105	-	-	380	268
	End	228	129	218	-	-	496	382
**Pfam:MutS_II (PF05188)**	Start	-	142	235	-	-	505	388
	End	-	284	361	-	-	676	542
**Pfam:GIY-YIG (PF01541)**	Start	1024	-	-	-	-	-	-
	End	1091	-	-	-	-	-	-
**TUDOR (PF00567)**	Start	-	-	-	-	-	121	-
	End	-	-	-	-	-	179	-

**Table 4 molecules-24-02493-t004:** Interactors of AtMSH proteins as retrieved by interactome analysis in STRING.

MSH	Interactor			CV
	Name	Accession	Function	
**AtMSH1**	MLH1	AT4G09140.1	MUTL-homologue 1; correcting IDLs in MMR coming from DNA replication, DNA damage or heterologous recombination in meiosis	0.999
	MLH3	AT4G35520.1	MUTL protein homolog 3; correcting IDLs in MMR coming from DNA replication, DNA damage or heterologous recombination in meiosis	0.996
	PMS1	AT4G02460.1	Postmeiotic segregation 1; correcting non-Watson–Crick base pairing and IDLs in MMR; coming from DNA replication, DNA damage or heterologous recombination in meiosis	0.999
	PCNA1	AT1G07370.1	Proliferating cellular nuclear antigen 1; auxiliary protein of DNA polδ; controls eukaryotic DNA replication	0.992
	PCNA2	AT2G29570.1	proliferating cell nuclear antigen 2; auxiliary protein of DNA polδ; controls eukaryotic DNA replication	0.991
	MSH2	AT3G18524.1	MUTS homolog 2; (see Introduction for detailed description)	0.981
	MSH5	AT3G20475.1	MUTS-homolog 5; (see Introduction for detailed description)	0.981
	RECA3	AT3G10140.1	RECA homolog 3; plays role in recombination ability DNA strand transfer	0.980
	TIL1	AT1G08260.1	TILTED 1; DNA polymerase II; involved in DNA replication. Important physiological role (timing and determination of cell fate during plant embryogenesis and root pole development; required for proper shoot (SAM) and root apical meristem (RAM) function; required for flowering repression	0.974
	TIL2	AT2G27120.1	TILTED 2; DNA polymerase II; involved in DNA replication, promotes cell cycle and cell type patterning. Contributes to flowering time repression	0.974
**AtMSH2**	MLH1	AT4G09140.1	MUTL-homologue 1; correcting IDLs in MMR coming from DNA replication, DNA damage or heterologous recombination in meiosis	0.999
	MLH3	AT4G35520.1	MUTL protein homolog 3; correcting IDLs in MMR coming from DNA replication, DNA damage or heterologous recombination in meiosis	0.997
	PMS1	AT4G02460.1	Postmeiotic segregation 1; correcting non-Watson–Crick base pairing and IDLs in MMR; coming from DNA replication, DNA damage or heterologous recombination in meiosis.	0.999
	PCNA1	AT1G07370.1	proliferating cellular nuclear antigen 1; auxiliary protein of DNA polδ; controls eukaryotic DNA replication	0.997
	PCNA2	AT2G29570.1	proliferating cell nuclear antigen 2; auxiliary protein of DNA polδ; controls eukaryotic DNA replication	0.998
	MSH7	AT3G24495.1	MUTS homolog 7; (see Introduction for detailed description)	0.997
	MSH6	AT4G02070.1	MUTS homolog 6; (see Introduction for detailed description)	0.983
	UVH1	AT5G41150.1	DNA repair endonuclease UVH1; probably involved in NER and repair of UV light damage, and oxidative damage. In vitro, repairs DSBs and is required for homologous recombination	0.991
	TIL1	AT1G08260.1	TILTED 1; DNA polymerase II; involved in DNA replication. Important physiological role (timing and determination of cell fate during plant embryogenesis and root pole development; required for proper shoot (SAM) and root apical meristem (RAM) function; required for flowering repression	0.974
	RECQSIM	AT5G27680.1	RECQ helicase SIM; Involved in DNA repair; 3′-5′ helicase specific for plants	0.991
	ERCC1	AT3G05210.1	DNA excision repair protein ERCC-1; involved in NER. In vitro, repairs DSBs and is required for homologous recombination. UVH1/RAD1-ERCC1/RAD10 complex acts as endonuclease	0.990
**AtMSH3**	MLH1	AT4G09140.1	MUTL-homologue 1; correcting IDLs in MMR coming from DNA replication, DNA damage or heterologous recombination in meiosis	0.999
	MLH3	AT4G35520.1	MUTL protein homolog 3; correcting IDLs in MMR coming from DNA replication, DNA damage or heterologous recombination in meiosis	0.997
	PMS1	AT4G02460.1	Postmeiotic segregation 1; correcting non-Watson-Crick base pairing and IDLs in MMR; coming from DNA replication, DNA damage or heterologous recombination in meiosis	0.999
	PCNA1	AT1G07370.1	proliferating cellular nuclear antigen 1; auxiliary protein of DNA polδ; controls eukaryotic DNA replication	0.955
	PCNA2	AT2G29570.1	proliferating cell nuclear antigen 2; auxiliary protein of DNA polδ; controls eukaryotic DNA replication	0.968
	AT2G02550	AT2G02550.2	PIN domain-containing protein; nuclease	0.965
	AT1G29630	AT1G29630.2	exonuclease 1; dsDNAexonuclease. May be involved in DNA mismatch repair (MMR)	0.965
	AT1G18090	AT1G18090.1	5′-3′ exonuclease family protein	0.965
	ERCC1	AT3G05210.1	DNA excision repair protein ERCC-1; involved in NER. In vitro, repairs DSBs and is required for homologous recombination. UVH1/RAD1-ERCC1/RAD10 complex acts as endonuclease	0.953
	RECQSIM	AT5G27680.1	RECQ helicase SIM; Involved in DNA repair; 3′-5′ helicase specific for plants	0.863
**AtMSH4**	MLH1	AT4G09140.1	MUTL-homologue 1; correcting IDLs in MMR coming from DNA replication, DNA damage or heterologous recombination in meiosis	0.999
	MLH3	AT4G35520.1	MUTL protein homolog 3; correcting IDLs in MMR coming from DNA replication, DNA damage or heterologous recombination in meiosis	0.999
	PMS1	AT4G02460.1	Postmeiotic segregation 1; correcting non-Watson–Crick base pairing and IDLs in MMR; coming from DNA replication, DNA damage or heterologous recombination in meiosis	0.998
	MSH5	AT3G20475.1	MUTS-homologue 5; (see Introduction for detailed description)	0.987
	RAD51	AT5G20850.1	DNA repair protein RAD51-like 1; binds ss- and dsDNA; DNA-dependent ATPase; repair of meiotic DBSs generated by AtSPO11-1 and in homologous recombination. Important for vegetative growth and root mitosis	0.984
	MUS81	AT4G30870.1	MMS andUV sensitive 81; part of endonuclease complex. Involved in DNA repair and homologous recombination (HR) in somatic cells.	0.980
	ATSPO11-1	AT3G13170.1	Meiotic recombination protein SPO11-1; part of meiotic recombination. Cleaves DNA to make DSB and start meiotic recombination	0.970
	RCK	AT3G27730.1	ROCK-N-ROLLERS; DNA helicase important for meiosis	0.964
	DMC1	AT3G22880.1	Disruption of meiotic control 1; May participate in meiotic recombination	0.958
	SPO11-2	AT1G63990.1	sporulation 11-2; involved in meiotic recombination. Cleaves DNA to make DSB and start meiotic recombination	0.931
**AtMSH5**	MLH1	AT4G09140.1	MUTL-homologue 1; correcting IDLs in MMR coming from DNA replication, DNA damage or heterologous recombination in meiosis	0.999
	MLH3	AT4G35520.1	MUTL protein homolog 3; correcting IDLs in MMR coming from DNA replication, DNA damage or heterologous recombination in meiosis	0.999
	PMS1	AT4G02460.1	Postmeiotic segregation 1; correcting non-Watson-Crick base pairing and IDLs in MMR; coming from DNA replication, DNA damage or heterologous recombination in meiosis	0.999
	PCNA1	AT1G07370.1	proliferating cellular nuclear antigen 1; auxiliary protein of DNA polδ; controls eukaryotic DNA replication	0.993
	PCNA2	AT2G29570.1	proliferating cell nuclear antigen 2; auxiliary protein of DNA polδ; controls eukaryotic DNA replication	0.993
	RECQ4A	AT1G10930.1	ATP-dependent DNA helicase Q-like 4A; DNA helicase possibly involved in repair of DNA	0.994
	RECQSIM	AT5G27680.1	RECQ helicase SIM; Involved in DNA repair; 3′-5′ helicase specific for plants	0.991
	RecQI3	AT4G35740.1	ATP-dependent DNA helicase Q-like 3; DNA helicase; possible role in DNA repair. Mediates DNA strand annealing	0.991
	RECQI1	AT3G05740.1	RECQ helicase l1; DNA helicase; possible role in DNA repair	0.991
	RECQ4B	AT1G60930.1	RECQ helicase L4B; DNA helicase; possible role in DNA repair; promotes crossovers	0.991
**AtMSH6**	MLH1	AT4G09140.1	MUTL-homologue 1; correcting IDLs in MMR coming from DNA replication, DNA damage or heterologous recombination in meiosis	0.999
	MLH3	AT4G35520.1	MUTL protein homolog 3; correcting IDLs in MMR coming from DNA replication, DNA damage or heterologous recombination in meiosis	0.998
	PMS1	AT4G02460.1	Postmeiotic segregation 1; correcting non-Watson–Crick base pairing and IDLs in MMR; coming from DNA replication, DNA damage or heterologous recombination in meiosis	0.999
	PCNA1	AT1G07370.1	proliferating cellular nuclear antigen 1; auxiliary protein of DNA polδ; controls eukaryotic DNA replication	0.995
	PCNA2	AT2G29570.1	proliferating cell nuclear antigen 2; auxiliary protein of DNA polδ; controls eukaryotic DNA replication	0.996
	MSH5	AT3G20475.1	MUTS-homologue 5; (see Introduction for detailed description)	0.984
	MSH2	AT3G18524.1	MUTS homolog 2; (see Introduction for detailed description)	0.983
	TIL1	AT1G08260.1	TILTED 1; DNA polymerase II; involved in DNA replication. Important physiological role (timing and determination of cell fate during plant embryogenesis and root pole development; required for proper shoot (SAM) and root apical meristem (RAM) function; required for flowering repression	0.974
	TIL2	AT2G27120.1	TILTED 2; DNA polymerase II; involved in DNA replication, promotes cell cycle and cell type patterning. Contributes to flowering time repression	0.974
	RECQSIM	AT5G27680.1	RECQ helicase SIM; Involved in DNA repair; 3′-5′ helicase specific for plants	0.970
**AtMSH7**	MLH1	AT4G09140.1	MUTL-homologue 1; correcting IDLs in MMR coming from DNA replication, DNA damage or heterologous recombination in meiosis.	0.999
	MLH3	AT4G35520.1	MUTL protein homolog 3; correcting IDLs in MMR coming from DNA replication, DNA damage or heterologous recombination in meiosis	0.997
	PMS1	AT4G02460.1	Postmeiotic segregation 1; correcting non-Watson–Crick base pairing and IDLs in MMR; coming from DNA replication, DNA damage or heterologous recombination in meiosis	0.999
	PCNA1	AT1G07370.1	proliferating cellular nuclear antigen 1; auxiliary protein of DNA polδ; controls eukaryotic DNA replication	0.996
	PCNA2	AT2G29570.1	proliferating cell nuclear antigen 2; auxiliary protein of DNA polδ; controls eukaryotic DNA replication	0.995
	MSH5	AT3G20475.1	MUTS-homologue 5; (see Introduction for detailed description)	0.984
	MSH2	AT3G18524.1	MUTS homolog 2; (see Introduction for detailed description)	0.997
	TIL1	AT1G08260.1	TILTED 1; DNA polymerase II; involved in DNA replication Important physiological role (timing and determination of cell fate during plant embryogenesis and root pole development; required for proper shoot (SAM) and root apical meristem (RAM) function; required for flowering repression	0.976
	TIL2	AT2G27120.1	TILTED 2; DNA polymerase II; involved in DNA replication, promotes cell cycle and cell type patterning. Contributes to flowering time repression	0.976
	RECQSIM	AT5G27680.1	RECQ helicase SIM; Involved in DNA repair; 3′-5′ helicase specific for plants	0.968

**Table 5 molecules-24-02493-t005:** Subcellular localization of AtMSH proteins.

Protein	Subcellular Localization	Subnuclear Localization
MSH1	Mitochondrion Chloroplast	- -
MSH2	Nucleus	Nucleolus
MSH3	Nucleus	Nucleolus
MSH4	Nucleus	Nucleolus
MSH5	Nucleus	Nucleolus
MSH6	Nucleus	Nucleolus
MSH7	Nucleus	Nucleolus

**Table 6 molecules-24-02493-t006:** AtMSH proteins accession numbersfrom the National Center for Biotechnology Information (NCBI) and the Arabidopsis Information Resource (TAIR).

Protein Name	Accession Number		Sequence Length
	NCBI	TAIR	
AtMSH1	Q84LK0.1	AT3G24320.1	1118 aa
AtMSH2	O24617.1	AT3G18524.1	937 aa
AtMSH3	O65607.2	AT4G25540.1	1081 aa
AtMSH4	F4JP48.1	AT4G17380.1	792 aa
AtMSH5	F4JEP5.1	AT3G20475.1	807 aa
AtMSH6	O04716.2	AT4G02070.1	1324 aa
AtMSH7	Q9SMV7.1	AT3G24495.1	1109 aa

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
