# Peer review of "Interactome Analysis and Docking Sites of MutS Homologs Reveal New Physiological Roles in Arabidopsis thaliana"

_molecules, 2019, doi:10.3390/molecules24132493_

Round 1

Reviewer 1 Report

Manuscript ID: molecules-513943

Entitled

Interactome analysis and docking sites of MutS homologs reveal new physiological roles in Arabidopsis thaliana.

The ms is and extensive data paper bases computer tool predictions of the mismatch repair proteins in Arabidopsis thaliana. The authors try to aim to predict new physiological roles based on interactome analysis from databases. A weak point is that the prediction study is not supported by any wet lab experiments/confirmations, although these prediction data has some interesting observations.

Minor

In this ms are the MSH like proteins discussed, it would be useful to include MLH and PMS proteins

Major

Page 1

Line 13,14 In order to fight off the stress..   . One …. repair pathway.

-These lines should less strong stated. The type of stress has to be announced.

-It is not clear if authors mean oxidative stress or DNA stress.

-The MMR is not an essential DNA repair pathway, different the DNA repair pathways HR, NHEJ (deleted in the Ku70 or Ku80 in humans) and BER dysfunction in these pathways is lethal. In humans the dysfunction/mutations in MMR proteins are associated in diseases lynch syndrome and may contribute to cause of huntington’s disease (the last is not clear). 

Line 14

Crusial … 

-Use a less strong wording, since MMR is not essential DNA repair pathway, it gives strong undertone.

Page 2

In humans the classical MMR proteins have not been identified to be present in the mitochondria, therefore careful wording is needed to address MSH1 present in mitochondria. 

 Page 7

It would be great if can be explained why almost all MSH proteins contain a nuclear as well as a nucleolus localization signal. While in humans only MSH6 is associated with nuclear and nucleolus localization (pmid number 20004149, Nuclear reorganization of DNA mismatch repair proteins in response to DNA damage). Would there be a functional explanation why these proteins would be in the nucleolus? Or is it just artifacts in the prediction tool?

Page 8

Table 4

As mentioned in the text MSH1 is restricted to mitochondria and chloroplast, however according to the table the MSH1 interacts with nuclear proteins including MSH2 which is only in the nucleus. Please explain? 

Table 4 is very extensive, it would be useful to remove function of the interacting proteins and move to the supplement since is seems that atMSH interact with several same proteins. This will make the table more compact and create a better overview.

Page 14

In eukaryotes humans/yeast it is shown that MMR proteins can facilitate other DNA repair pathways in the repair of DNA damage or during processes metabolic processes such as class switching. 

It is an overstatement to if the atMSH proteins interact with other DNA repair proteins that those are part of the pathway as is stated that MSH2/3 function in HR. DNA repair proteins are flexible and facilitate other DNA repair pathways, but not make part of the conical DNA repair pathways as HR, NHEJ, NER or BER. 

It is suggested to read the follow, pmid number 28927527, DNA mismatch repair and its many roles in eukaryotic cells.

Since the study is based on prediction conclusion/discussion has to be stated carefully.

Author Response

Dear Reviewer

Thank you for you your valuable comments which are making this paper much more clear and sound. Please find all answers to your comments in red font in the manuscript file. I will be available for any further inquires about MSH paper.

Please find attached the corrected manuscript, answers to your comments and cNLS of MSH in the end of this file

Best Regards,

Prof. Dr. Mohamed Ragab Abdel Gawwad, BSc, MSc, PhD

Genetics & Bioengineering 

Faculty of Engineering and Natural Sciences

mragab@ius.edu.ba

Office : 00387 33 957 203

Fax : 00387 33 957 105

International University of Sarajevo 
Hrasnička cesta br. 15, 71210 Ilidža - Sarajevo, BiH

https://orcid.org/0000-0002-8115-812X

https://www.researchgate.net/profile/Mohamed_Abdel_Gawwad/research

Reviewer 2 Report

The manuscript " Interactome Analysis and Docking Sites of MutS Homologs 3 Reveal new Physiological Roles in Arabidopsis Thaliana " by Mohamed Ragab AbdelGawwad and Aida Marić frepresent an interesting study of the structure of MSH proteins, which are involved in DNA mismatch repair. Authors predict the possible interactors, based on the protein modelling.

This is a pure bioinformatic study, which however could be of interest for broad community of plant researchers. Thus, I would recommend to make it clearer to biologist lacking the bioinformatic expertise.

Comments:

1. The introduction is focused on the formation of homo-heterodimers betwen individual MSH proteins, but since the study further focuses on the general  MSH interactome, some described interactions - in Arabidopsis if there are - or other models should be mentioned.

2. The Figure and Table legends are not at all extended, please provide full figure legends, explaining what is shown in figures and tables 

a)  the Fig.2. legend seems wrong; please correct also the labelling of individual images to have same letter sizes, same format etc. 

b) Tables 3 (and 4) is not displayed well, thus MutS I and Muts II domains appear confusing; also the text describing the data lack the MutS I domain

c) line 141 I suggest to specify, that TUDOR functions are not described in plants. Also, following sentences lack the reference, and it should be described more extensively. Number of references can be track for Arabidopsis, e.g. https://www.ncbi.nlm.nih.gov/pmc/articles/PMC4634149/

d) I suggest to make a Venn diagram sumarising the interacting proteins 

The interactome prediction was done also in some other studies (e.g. http://www.plantphysiol.org/content/145/2/317), it should be mentioned and the used tools discussed, also, are there any other tool to predict the interactome to be compared to currently used one? 

 3. In materials and methods some bioinformatic tools are not well described, please add description how they work. 

line 285: word indeed is perhaps not needed

4. Discussion

Authors should avoid the word "function", since this is not a functional study, but prediction based. Again, if there are some in vivo described interactions,there should be discussed in relation to obtained predicted data, and it should be clear from the  text if authors talk about the prediction based data or in vivo confirmed data. 

Author Response

Dear Reviewer

Thank you for you your valuable comments which are making this paper much more clear and sound. Please find all answers to your comments in blue font in the manuscript file. I will be available for any further inquires about MSH paper.

Best Regards,

Prof. Dr. Mohamed Ragab Abdel Gawwad, BSc, MSc, PhD

Genetics & Bioengineering 

Faculty of Engineering and Natural Sciences

mragab@ius.edu.ba

Office : 00387 33 957 203

Fax : 00387 33 957 105

International University of Sarajevo 
Hrasnička cesta br. 15, 71210 Ilidža - Sarajevo, BiH

https://orcid.org/0000-0002-8115-812X

https://www.researchgate.net/profile/Mohamed_Abdel_Gawwad/resear

Round 2

Reviewer 1 Report

Page 1 

Spacing in the new added sentence at the start and end.

Change the follow; ……. free radicals, “replication errors, polymerase slippage” and chemical mutagens ……

Next sentence change

In order to face “ the base oxidation or DNA replication stress”, plants have developed …

Page 7

The section added in, in blue, has to be carefully checked for grammar errors.

Author Response

Dear Reviewer

Thank you for your valuable comments. please find the corrections below and in the manuscript.

Page 1 

Spacing in the new added sentence at the start and end.

It is done and Highlighted in yellow color

Change the follow; ……. free radicals,“replication errors, polymerase slippage”  and chemical mutagens ……

Next sentence change

In order to face “ the base oxidation or DNA replication stress”, plants have developed …

 It is done and Highlighted in green color

Page 7

The section added in, in blue, has to be carefully checked for grammar errors.

It is done and Highlighted in yellow color

Thank you in advance

Best Regards,

Dr. Mohamed Ragab Abdel Gawwad, BSc, MSc, PhD

Associate Professor

Genetics & Bioengineering 

Faculty of Engineering and Natural Sciences

mragab@ius.edu.ba

Office : 00387 33 957 203

Fax : 00387 33 957 105

International University of Sarajevo 
Hrasnička cesta br. 15, 71210 Ilidža - Sarajevo, BiH
